METHODS

# Virus-host interactions predictor (VHIP): Machine learning approach to resolve microbial virus-host interaction networks

**G. Eric Bastien**, **Rachel N. Cable**, **Cecelia Batterbee**, **A. J. Wing**, **Luis Zaman***, **Melissa B. Duhaime***

Department of Ecology and Evolutionary Biology, University of Michigan, Ann Arbor, Michigan, United States of America

* zamanlh@umich.edu (LZ); duhaimem@umich.edu (MBD)

**Data Availability Statement:** All relevant data are within the manuscript and its Supporting Information files. Code written for analyses and

## Abstract

Viruses of microbes are ubiquitous biological entities that reprogram their hosts' metabolisms during infection in order to produce viral progeny, impacting the ecology and evolution of microbiomes with broad implications for human and environmental health. Advances in genome sequencing have led to the discovery of millions of novel viruses and an appreciation for the great diversity of viruses on Earth. Yet, with knowledge of only *"who is there?"* we fall short in our ability to infer the impacts of viruses on microbes at population, community, and ecosystem-scales. To do this, we need a more explicit understanding *"who do they infect?"* Here, we developed a novel machine learning model (ML), Virus-Host Interaction Predictor (VHIP), to predict virus-host interactions (infection/non-infection) from input virus and host genomes. This ML model was trained and tested on a high-value manually curated set of 8849 virus-host pairs and their corresponding sequence data. The resulting dataset, 'Virus Host Range network' (VHRnet), is core to VHIP functionality. Each data point that underlies the VHIP training and testing represents a lab-tested virus-host pair in VHRnet, from which meaningful signals of viral adaptation to host were computed from genomic sequences. VHIP departs from existing virus-host prediction models in its ability to predict multiple interactions rather than predicting a single most likely host or host clade. As a result, VHIP is able to infer the complexity of virus-host networks in natural systems. VHIP has an 87.8% accuracy rate at predicting interactions between virus-host pairs at the species level and can be applied to novel viral and host population genomes reconstructed from metagenomic datasets.

## Author summary

The ecology and evolution of microbial communities are deeply influenced by viruses. Metagenomics analysis, the non-targeted sequencing of community genomes, has led to the discovery of millions of novel viruses. Yet, through the sequencing process, only DNA sequences are recovered, begging the question: which microbial hosts do those novel viruses infect? To address this question, we developed a computational tool to allow

figures generated as part of this manuscript is made available on Github (https://github.com/DuhaimeLab/VHIP_analyses_Bastien_et_al_2023) The tool VHIP, described in the manuscript, is made available as a Python package through conda-forge and PyPI. The source code is made public on Github (https://github.com/DuhaimeLab/VirusHostInteractionPredictor).

**Funding:** This study is based upon work supported by the National Science Foundation under Grant No. 2055455 awarded to MBD and LZ and 1813069 awarded to LZ and by funding to MBD through the National Oceanic and Atmospheric Administration Great Lakes Omics program distributed through the University of Michigan Cooperative Institute for Great Lakes Research NA17OAR4320152. This is CIGLR contribution number 1250. The funders had no role in study design, data collection and analysis, decision to publish, or preparation of the manuscript.

**Competing interests:** The authors have declared that no competing interests exist.

researchers to predict virus-host interactions from such sequence data. The power of this tool is its use of a high-value, manually curated set of 8849 lab-verified virus-host pairs and their corresponding sequence data. For each pair, we computed signals of coevolution to use as the predictive features in a machine learning model designed to predict interactions between viruses and hosts. The resulting model, Virus-Host Interaction Predictor (VHIP), has an accuracy of 87.8% and can be applied to novel viral and host genomes reconstructed from metagenomic datasets. Because the model considers all possible virus-host pairs, it can resolve complete virus-host interaction networks and supports a new avenue to apply network thinking to viral ecology.

## Introduction

The development of metagenomic sequence analyses has led to unprecedented discoveries in microbial science [1,2], owing to the ability to study viruses and cellular microbes in their quintessential contexts. In particular, metagenomics has shed light on the genomic diversity and ubiquity of viruses [3–7]. Those viral populations are reconstructed from metagenomic data and identified as novel based on their sequence similarity to known viruses, expanding the total number of distinct uncultured viruses to millions the last decade alone. [3,8–11]. With these discoveries, there is mounting interest in characterizing how viruses impact microbial communities [12]. During infection, viruses influence ecological processes at multiple scales by shaping host population dynamics [13,14], modulating horizontal gene transfer [15,16], and reprogramming host metabolic pathways that can modulate the flux of environmental nutrients [17–20]. Given the central role viruses play in the ecology and evolution of their microbial hosts, there is great interest in including them in biogeochemical modeling by leveraging global-scale metagenomic data. While the population genomes of viruses and their microbial hosts can be reconstructed from metagenomic data with increasing accuracy, the inclusion of viruses in community metabolic models that can one day be scaled to ecosystems is impeded by the absence of arguably the most important question about any novel virus: who does it infect [11,21,22]?

To bridge this knowledge gap, various approaches have been explored. Phylogeography-based approaches have been successful in predicting virus-host associations in eukaryotic systems [23–25]. However, this approach is not applicable for viruses infecting microbes, as microbes show weak patterns of biogeography [26]. Another method leverages patterns of coevolution that can be extracted from the genomic sequences of the viruses and their host to predict the most likely taxa that can be infected by a given virus [27–31]. This approach of leveraging genomic signals to predict virus-host association is possible because viruses rely on their host machinery to complete their life cycle and evolve to better utilize those resources by matching the codon biases of their host [20,32,33]. This results in a meaningful and capturable signal that is embedded in the sequences of the viruses and their known host [34,35] Those host prediction tools (HPTs) are, however, limited in scope. They typically only allow viral sequences as input, which restricts testing to host sequences that already exist in pre-defined reference databases, or they require sufficient expertise and resources from users to re-train the models to include new hosts sequences. These limitations restrict the applicability of such tools, as they are difficult to use to study viral host range in natural community contexts with newly discovered host populations. In addition, they typically focus on predicting the most likely taxa a virus infects, which does not reflect the breadth of natural virus-host range profiles [36,37] that can span different taxonomic levels [37–40]. Further, as they do not predict non-

infection interactions, that is, a virus's inability to infect a host, existing HPTs cannot be used to resolve virus-host interaction networks.

To address these limitations, we collected lab-verified viral-host interactions (infection and non-infection data) from public databases and literature and compiled them into a single dataset named 'Virus Host Range network' (VHRnet). This data is essential for exploring the strengths of the virus-host coevolution signals, assessing existing HPTs, and for developing a novel model that can predict both infection and non-infection relationships for all pairwise sets of viruses and putative hosts the prediction model may someday encounter. In this study the VHRnet data were used to (1) quantify and evaluate genome-derived signals of coevolution captured in lab-validated virus-host pairs, (2) develop a machine learning model that leverages these virus-microbe coevolutionary signals, and (3) assess the accuracy of the model in predicting interactions (i.e., infection or non-infection) at the species level. The resulting model developed and described here is named VHIP for Virus-Host Interaction Predictor.

## Results

### VHRnet, a manually curated host range dataset unparalleled in size and scope

To train and test machine learning model approaches to predict virus-host relationships, we aimed to compile the most comprehensive host range dataset available, wherein all viral and host genome sequences are also publicly available (Fig 1A). For this, we relied on the fact that every virus in the NCBI RefSeq database has a host associated with it via the '/lab_host'tag in the GenBank file controlled syntax. In addition, for each virus in the RefSeq database, we searched published studies beyond the genome reports for documented host range trials (S1 Table). A total of 8849 lab-tested interactions were collected and compiled (Fig 2A). The majority of interactions in this dataset were non-infection (n = 6079), owing to a small number of large-scale host range studies that diligently tested and reported all virus-host pairs in their study (Fig 2B), rather than only reporting cases of infection, as is done in the vast majority of virus-host studies (S1 Table). This resulting dataset was named VHRnet (S2 Table), for Virus-Host Range network. It comprises 375 unique host species and 2292 unique viruses. The

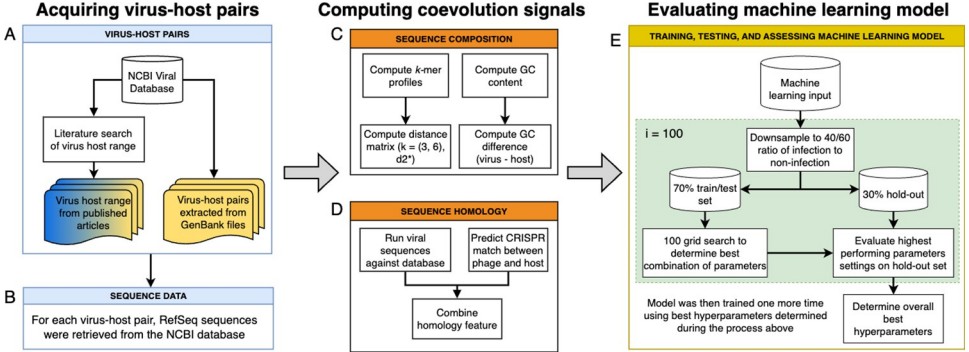

**Fig 1. Design of data collection, features computation, and evaluation of machine learning approach developed in this study A.** Metadata describing virus-host pairs was retrieved and compiled from NCBI and literature. Blue and yellow color indicates provenance of infection versus non-infection information, i.e., non-infection data came only from literature studies. **B.** Sequences of each virus and host were retrieved. Signals of coevolution were calculated for each virus-host pair, including C. sequence composition and D. sequence homology. E. The virus-host pairs and their computed signals of coevolution were used to train VHIP. To determine the best parameters for the machine learning model, a grid search was performed and bootstrapped 100 times on a training/testing set and evaluated on a hold-out set. The model was then retrained for a final time using the best hyperparameters.

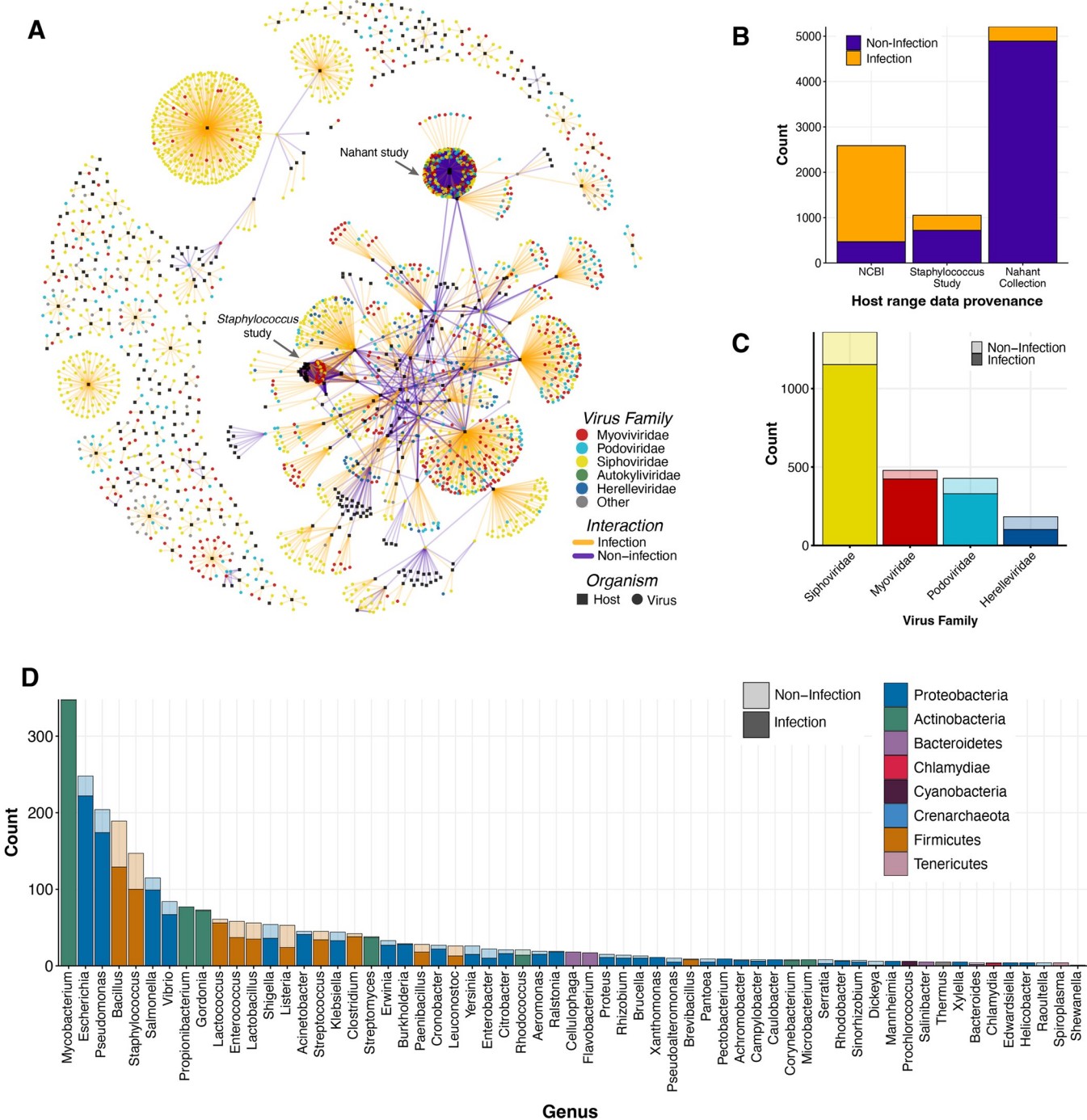

**Fig 2. VHRnet network visualization and content characteristics. A.** Network visualization of VHRnet, where an edge connects a viral node to a host node if that pair has been experimentally tested. The edge is colored by the interaction class (infection/non-infection) and the nodes by virus taxonomy or whether it is a host node (black squares are host species). **B.** Origin of known lab-tested infection and non-infection data across dataset sources. **C.** Distribution of family classifications for subset of viruses in VHRnet collected from NCBI. Lighter transparency represents the proportion of non-infection reports by viral family, relative to the solid portion, which represents known infection reports by viral family. **D.** Distribution of host genera for subset of hosts in VHRnet collected from NCBI. Lighter color transparency represents the proportion of non-infection relative to infection (solid color).

majority (94.7%) of the viruses belong to the Caudovirales order (Figs 2C and S1), which may be driven by culture techniques biases [36,41]. There are biases in the host taxa represented as well, with the majority of the tested hosts human pathogens (Figs 2D and S2), partially driven in the recent surge of phage therapy research [42].

The majority of the viruses (n = 1962, 84.2%) were reportedly tested against a single host; these pairings came from the '/lab_host'tag of their NCBI GenBank files. Of the remaining viruses (n = 369, 18.8%), most (81.8%) were tested against different host species with no cross-genus tests. The percentages of viruses tested across clades gets increasingly smaller at higher taxonomic levels: 78.5% were tested at most across families, 77.5% were tested across order, 9.2% were tested across class, and 0% were tested across phyla or domain (S3 Fig). However it is important to note that the numbers of cross-clade host testing were heavily influenced by two large host-range scale studies (herein the '*Staphylococcus* study' [40] and 'Nahant study' [36]), in which every potential virus and host pair was experimentally tested, resulting in a "complete" virus-host network (Fig 2A and 2B) [36,40]. 70.8% of the VHRnet pairs came from those two studies alone. When excluding those two large host range studies, only 82 viruses in VHRnet (4.01%) were tested against multiple species. Out of those viruses, 65.6% were tested against hosts belonging to different genera, 52.4% were tested against different families, 46.3% against different order, 41.2% against different class, and 0 viruses were tested against different hosts belonging to different phyla.

## VHRnet provides opportunity for holistic comparison of Host Prediction Tools (HPTs)

The diligent lab-testing of all virus-host pairs from *Staphylococcus* and Nahant studies (Fig 2A and 2B) presented an opportunity to directly compare existing HPTs (S3 Table) on the same lab-validated infection/non-infection data. While HPT benchmarking is common, such a direct comparison of HPTs has been challenging given that each tool uses similar but not identical datasets for model testing and training. For the most objective assessment, testing datasets must exclude training data, which is not possible when part of the data used to compare outputs of one model was included in the training sets of other benchmarked models. In assessing the accuracy of HPTs against the Nahant Collection and *Staphylococcus* datasets in this study, there were three possible outcomes: (1) correct host was predicted, (2) incorrect host was predicted, (3) a host was predicted that was not experimentally tested ("untested", Fig 3). Note that if there are multiple known hosts for a virus, these models only need to predict one correct host to obtain 100% accuracy in this evaluation.

We found that the accuracy of each HPT decreased as we considered more resolved host taxonomic levels (i.e., from domain to species level). Furthermore, the number of predicted virus-host pairs that were not experimentally tested also increased with increasing host taxonomic resolution (Fig 3). Existing HPTs performed better in predicting hosts for viruses in the *Staphylococcus* study than for viruses in the Nahant study, with overall fewer wrong predictions. Regardless, existing HPTs do not have the resolution to reliably predict species level virus-host interactions. Moreover, due to the limitation that stems from the reference data used by HPTs (that is, only one known host per virus), these tools typically deliver only their highest scoring host prediction as output, which can lead to uncertainty in output data interpretation and downstream use. For instance, if there are five total predictions, all of low quality scores, the best of the poor predictions will be reported as the "most likely host", rather than returning a prediction of "no infection". Similarly, if five hosts are predicted with similarly high scores, only a single prediction is chosen, rather than five "infection" predictions. This forced one-to-one prediction output model design limits the use of current HPTs and does not

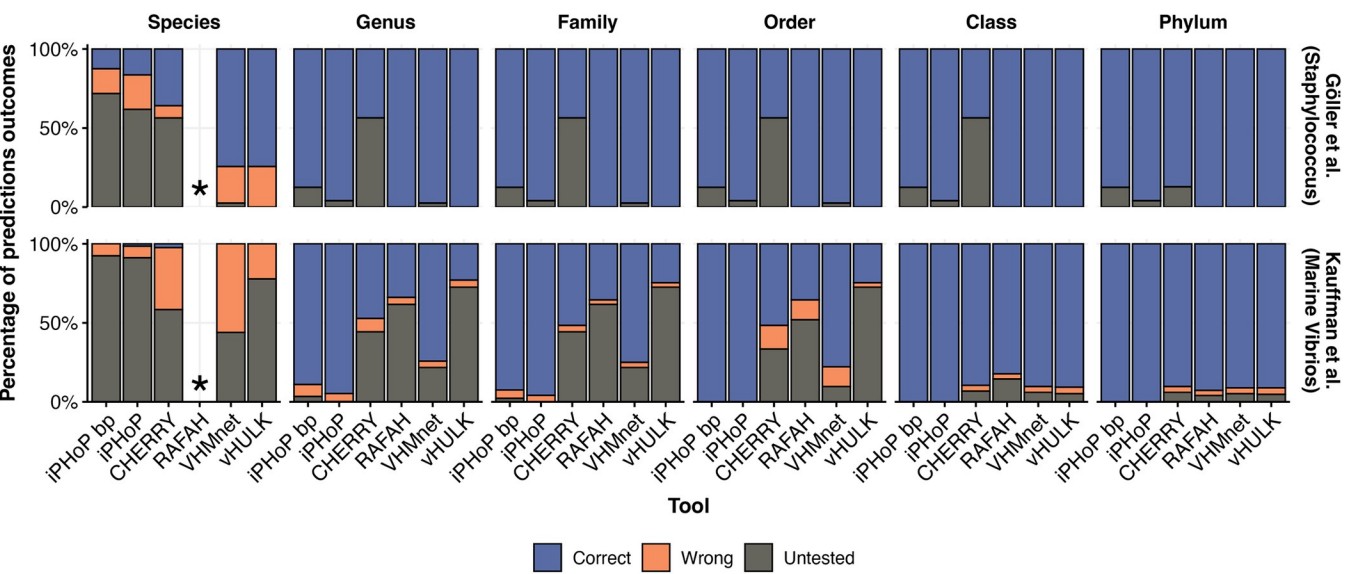

**Fig 3. Evaluation of the accuracy of existing virus-host predictions models using two complete the *Staphylococcus* (40) and Nahant (36) studies.** Accuracy is evaluated at different host taxa thresholds, from species to phylum. The relative proportion of correct, incorrect, and untested predictions is represented by bar color. Untested predictions are hosts predicted by an HPT but that are not experimentally validated as correct or incorrect. The prediction that received the highest score by each HPT was evaluated. Because iPHoP outputs multiple predictions, both the highest scoring prediction ("iPHoP bp") and the full set of predictions ("iPHoP") were reported. *RAFAH does not return predictions at the species level.

reflect the dynamics of virus-host relationships. Motivated to move beyond the one-to-one output design, we next leveraged the VHRnet data to identify and quantify genome-derived signals of coevolution between known infection pairs (relative to non-infection pairs) at the virus and host species level. We hypothesized that these data could be used to develop a many-to-many HPT design.

Genomic signals of virus adaptation to their host(s) are discernible at the species level Signals of coevolution were computed for each virus-host pair in VHRnet (Fig 1A and 1B), which can be broadly divided into two categories: sequence homology (Fig 1C) and sequence composition (Fig 1D) [41]. For sequence homology, virus and host genomes were scanned for stretches of DNA sequence homology, which may indicate past virus-host interactions, such as CRISPR activity or horizontal gene transfer (HGT). We first identified bacteria and archaea in the VHRnet dataset with predicted CRISPR-Cas systems, then retrieved their associated CRISPR spacer motifs, which are short DNA sequences acquired from previous exposure to a foreign genetic element as part of the CRISPR-Cas adaptive immune response. Viral genomes were then scanned for the presence of these identical motifs. Stretches of non-CRISPR sequence homology, which we attributed to HGT events, between viruses and their putative host sequences were similarly identified between VHRnet viruses and hosts, with the difference being their search was not restricted to predicted CRISPR spacer motifs (minimum identity percentage 80 with minimum hit length 500 bp). Across the VHRnet data, instances of shared sequence homology between viral and host sequences were rare: sequence matches to CRISPR spacers were identified in 184 viruses out of 2292 (8.03%) and sequence matches attributed to HGT were identified in 340 viruses (14.83%). That the frequency of CRISPR matches was low was not surprising given that an estimated 50% of bacteria and 10% of archaea lack CRISPR-Cas viral defense systems [43].

In addition to sequence homology, we quantitatively evaluated signals of virus-host coevolution based on sequence composition. Because viruses rely on their host machinery to

complete their life cycle [44], their genomes have a tendency towards matching the nucleotide usage biases of their hosts over coevolutionary time (the process of 'genome amelioration') [45–47]. We computed $k$-mer profiles at $k = 3$, 6, and 9 for all viral and host sequences in VHRnet. We used both the Euclidean and the d2* distance metrics [35] to compute $k$-mer profile similarities between viruses and hosts and then to evaluate which measure encoded the strongest signals for virus-host predictions given our study design. As previously reported [35], the $d_2$* metric outperformed the Euclidean distance metric in its ability to resolve sequence composition-based signals of virus-host coevolution (S4 Fig). In other applications, $d_2$* has been shown to predict viral hosts at the genus level with twice the accuracy of the Euclidean metric [35]. The $d_2$* metric differs from other distance metrics as it takes into consideration the background oligonucleotide patterns of the sequences being compared [48]. While the Euclidean metric remains the most popular for binning metagenomic contigs in the reconstruction of microbial populations from metagenomes, our results suggest that the $d_2$* may be a better metric for binning as well, especially with continued optimization to reduce its computational burden. The $d_2$* algorithm is rewritten here as a Python package to ease accessibility.

As longer $k$-mers ("words") are considered, the number of possible words increases exponentially, e.g., 4 possible letters raised to the power of $k$ means that for a 7 character sequence stretch [47], 16,384 words are possible. The length of the sequence must be sufficiently long, such that the frequency of each word is likely to be detected within the virus and host genomes being considered. If the $k$-mer is too long relative to the genome length, zeroes accrue in the $k$-mer frequencies table, which leads to spurious distance values, a behavior that has been previously posited by Edwards *et al*, 2016 [41]. Here we assessed the impact of this behavior on prediction performance, given that we could now explicitly compare $k$-mer profile distances of known infection and known non-infection cases. Although the signal for certain virus-host pairs may get stronger at higher $k$-lengths, this is not a universal trend [35] (Fig 4A). At $k = 9$, some known non-infection pairs have smaller distances to known infection pairs (Fig 4A, orange arrow), regardless of the distance metric used. This does not happen when using k-lengths of 3 or 6. Thus we do not recommend using $k$-mers larger than 6 for purposes of virus-host predictions.

While it is commonly recognized that the %G+C contents of viruses are typically very similar to those of their hosts [49], the %G+C differences between viruses and hosts (%G+C of virus—%G+C of host) have also been previously recognized as a feature that could provide valuable information for virus-host predictions [41]. As %G+C difference is not currently considered by existing HPTs, we evaluated the potential of this signal to capture coevolutionary relationships. A decades-old study of 59 virus-host associations reported that viruses were on average 4% richer in AT [49]. This trend was attributed to the fact that G and C are energetically more expensive nucleotides to synthesize than A and T, that ATP is an abundant cellular molecule and more readily available for genome incorporation, and that there are more diverse pathways (and fewer metabolic bottlenecks) to synthesize A and T, as compared to, e.g., C [49]. In our set of 8849 virus-host interactions in VHRnet, we found a remarkably consistent result: viruses are on average 3.5% AT-richer relative to their host (Fig 4B, horizontal boxplot along top). We found that while the %G+C difference of a majority (68%) of the virus-host pairs fall within a narrow range of -4% to 4% (Fig 4C), the overall range of %G+C differences was quite broad, ranging from -40% to 32% (Fig 4C). These trends support that both the magnitude and direction of %G+C difference may be distinguishing features of virus-host infection pairs.

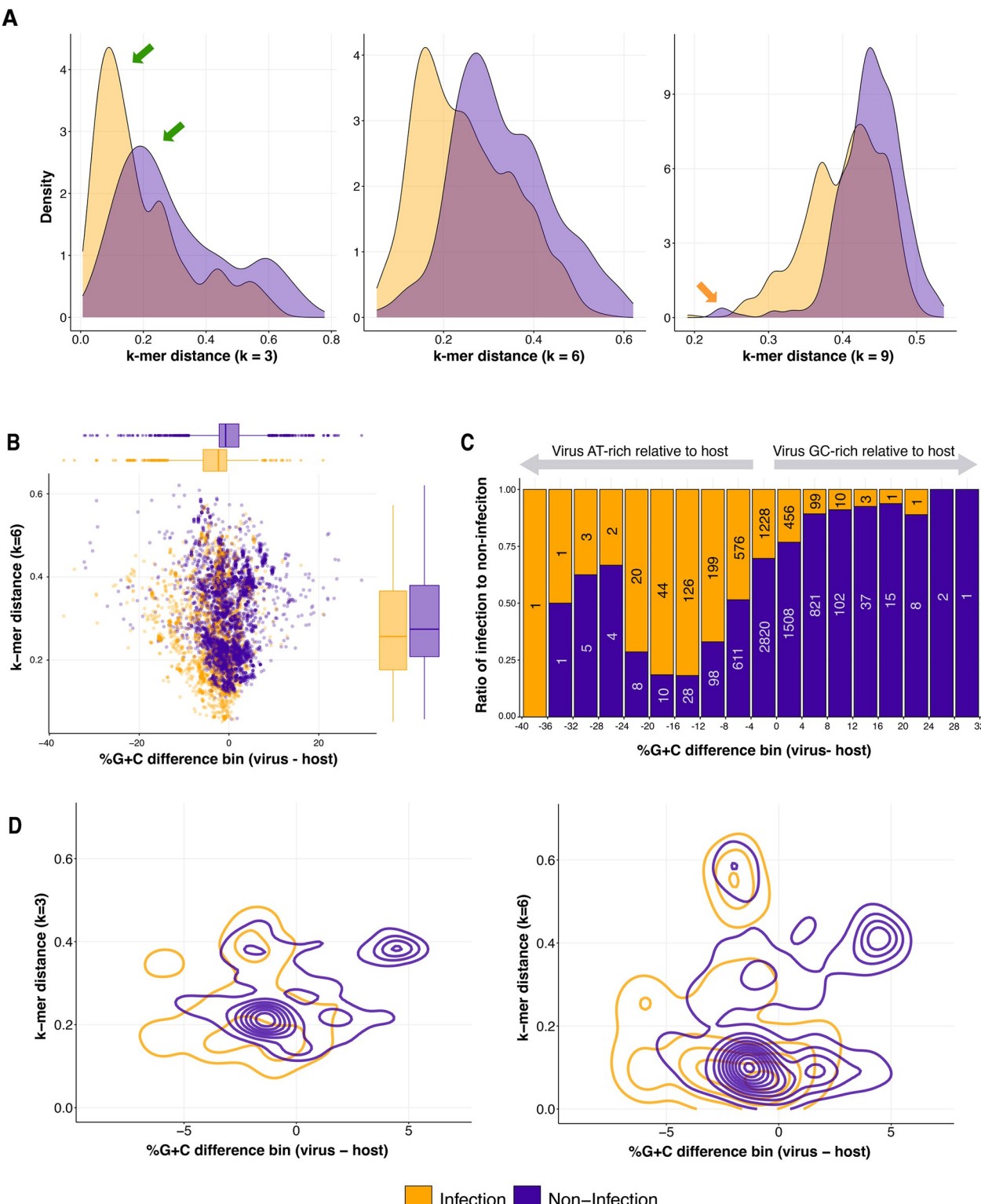

**Fig 4. Comparison of coevolution signals captured at the species taxonomic level. A.** Density plots of k-mer distances using k-length of 3, 6, and 9 between viruses and their hosts, colored by known interaction. **B.** Dot plot of %G+C difference bin against *k*-mer distance using d2*, colored by known interaction. Box and whisker plots represent the distribution values for the x-axis and y-axis, colored by known interaction (middle bar is median, quartile shows 25th and 75th percentile, and whiskers shows 1.5 times interquartile range). **C.** Stacked bar plot showing the ratio of lab-verified infection to non-infection for %G+C difference bins. Numbers inside the bars note the number of events observed for each %G+C

difference bin. Note that this number decreases sharply towards the extremes. **D.** Density plot of %G+C difference and *k*-mer distance between viruses and lab-tested hosts, colored by known interaction. Left plot is with k set to 3 and the right plot is with k set to 6.

### VHIP, a machine learning-based model, leverages the VHRnet dataset to predict virus-host interactions

Machine-learning model approaches are well-suited for capturing relationships between data features and have been applied to leverage genome-derived signals of virus-host coevolution using limited host range data in previous studies [50,51]. We sought to leverage VHRnet virus-host sequence pairs and the features we identified to assess, develop, and evaluate the performance of different machine learning models for virus-host predictions. Inclusion of infection and non-infection data points in VHRnet allows us to train a model predicting either infection or non-infection without any assumptions about virus host range. Before training machine learning-based models, we evaluated the Pearson pairwise correlations between the virus-host genomic signals (features used for machine-learning approach) (S5 Fig, and S4 Table) to determine whether any of the features are strongly correlated. Strongly correlated features typically do not bring additional information for prediction and would increase the complexity of the ML model, which is generally avoided when designing sound ML models [52,53]. Of all our evaluated features, only the k-3 and k-6 features were strongly correlated (Pearson = 0.94, S4 Table). However, when comparing the k-3 distance against %G+C difference and k-6 distance against %G+C difference, different patterns emerged (Fig 4D), suggesting that both the *k*-3 distance and *k*-6 distance encode different signals that can be leveraged by machine learning model approaches.

For a single feature to be strong enough to serve as a singular predictor of infection, there would need to be no overlap between infection and non-infection data points. This was not observed. For every feature, there was an overlap in the infection and non-infection density plots (Figs 4A and S3 diagonal plots). This strategy identifies regions of feature overlap and regions of distinction (i.e., valuable non-redundant information encoded in the features; green arrows Fig 4A) was also used to evaluate the power of combining features, such as the decision above to keep both k-3 and k-6 distances in the model (Fig 4D). Further, while virus-host genome amelioration was observed in the *k*-mer frequencies (resulting in low virus-host *k*-mer distances), we observed that viruses also have a strong tendency to remain AT-rich relative to their hosts (Fig 4B), consistent with prior reports [49].

We next evaluated the performance of different machine learning algorithms given our selected features (%G+C difference, k-3 distance, k-6 distance, homology hits). The Gradient Boosting Classifier and Random Forest performed best out of the classifiers we tested, with an average accuracy of 87.5% and 88.3% respectively (S6 Fig). However, the Gradient Boosting Classifier was used rather than the Random Forest, as it achieved comparable results with shallower trees, meaning fewer decision nodes were needed to reach comparable prediction performance. This is considered best practice for yielding the highest accuracy, while not overfitting the model [52]. The best performing model was then assessed on the hold-out dataset. We bootstrapped this approach 100 times and tracked which set of hyperparameters yielded the highest accuracy on the hold-out set for each iteration (Fig 1E). Across the 100 best models from each bootstrapping iteration, the average area under the receiver operator characteristic (AUROC) value was 0.93 ± 0.004 (S7 Fig). AUROC is a metric used to evaluate the accuracy of classification models, whereby a value of 1 represents a model with perfect predictions, a 0 represents only incorrect predictions, and 0.5 represents a model that makes random predictions.

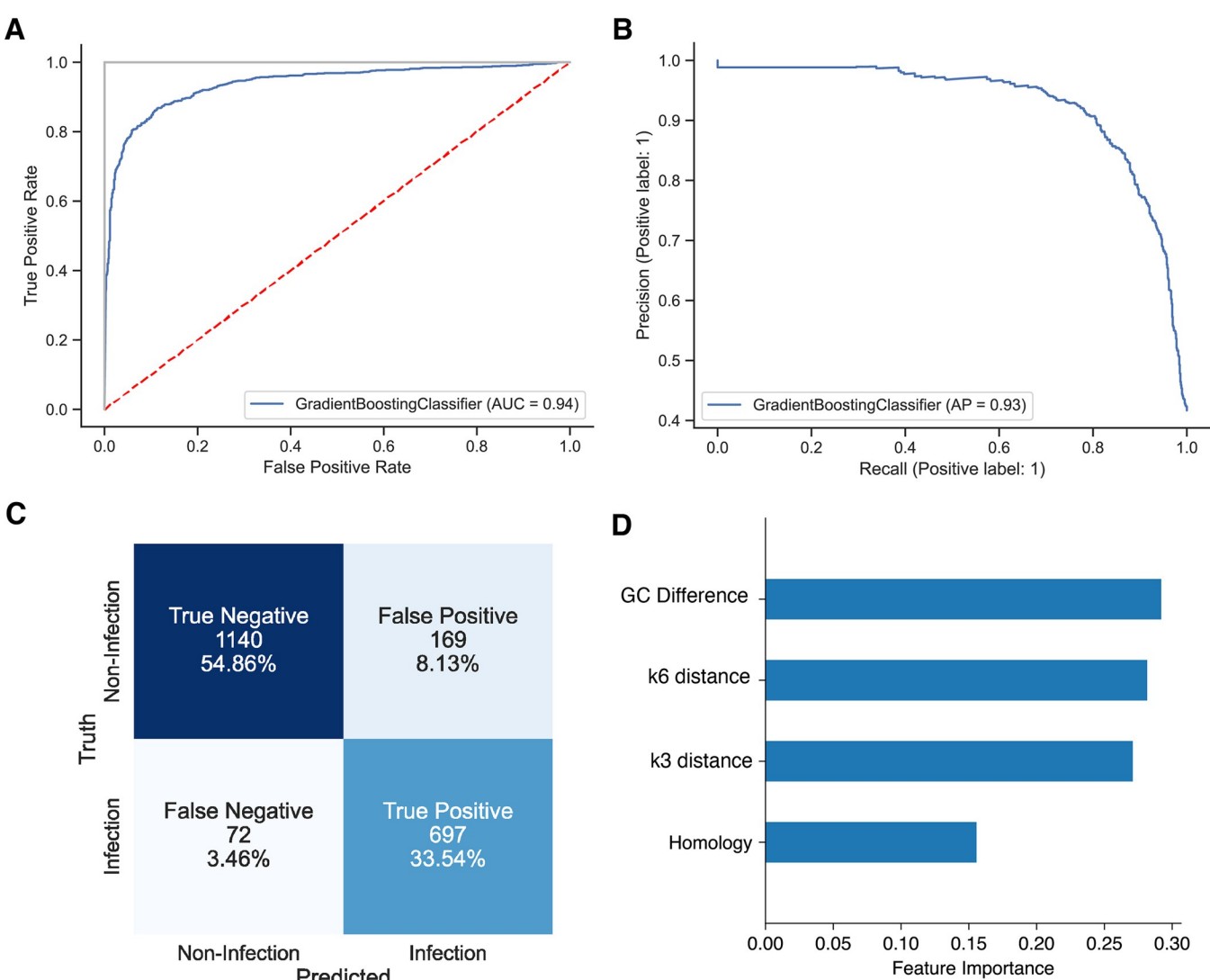

**Fig 5. Model performance and feature evaluation. A.** ROC curve of final trained Gradient Boosting Classifier. Red dashed line represents a model that makes random predictions, whereas the gray line represents a model that has 100% accuracy. The blue line is the behavior for VHIP. **B.** F1 curve of the final trained Gradient Boosting Classifier. **C.** Confusion matrix showing all possible outcomes, counts, and percentages. **D.** Relative importance of each feature that VHIP used to predict virus-host interactions.

We trained the machine learning model a final time using the entire dataset and the best hyperparameters determined from the performance analysis. The model can predict virus-host species interactions with 87.8% accuracy, defined as the number of correct predictions divided by total number of predictions on the test set. To assess VHIP performance, we considered all possible prediction outcomes: true positive (TP), true negative (TN), false positive (FP), and false negative (FN). False positives are commonly known as Type I error, and it represents cases when the model predicted infection but the virus-host pair is a case of non-infection. False negatives are commonly known as Type II errors, where the model predicted non-infection when it should have predicted infection. The AUROC score (the area under the receiver operating characteristics) for VHIP is 0.94 (Fig 5A); this is a measurement between the false positive rate (Type I error) against the true positive rate (model accurately predicting infection). The F1 score for VHIP is 0.93 (Fig 5B), which is the harmonic mean of the precision

(number of true positive results divided by the number of all positive results, TP / (TP + FP))
and recall (number of true positive results divided by the number of all samples that should
have been identified as positive (TP / (TP + FN)). The F1 score ignores the true negative and
may be misleading for unbalanced classes [54]. Finally, we computed Matthew's Coefficient
Correlation (MCC), which considers all four possible outcomes (TP, TN, FP, and FN)
(Fig 5C), and VHIP's scored 0.75 (where 1 represents a model that is perfectly accurate and 0 a
model that is making random predictions).

Interestingly, all the features used by the model contain information that can be leveraged
for predictions, but based on feature importance determined by the model during the training
phase, sequence composition features are the most useful for virus-host predictions (Fig 5D),
with %G+C difference encoding the strongest signal the model leverages. Note that because
the presence of sequence homology matches between viruses and hosts are rare, we combined
instances of homology between viral genomes and CRISPR spacers and between viral genomes
and putative host genomes into a single feature termed 'homology'. The model is available
through conda-forge and PyPI. The source code is available on Github.

To assess the effect of data provenance on model performance, we trained a machine learn-
ing model on a subsampled dataset containing 3159 data points but containing an equal
amount from each source (i.e., NCBI, Nahant Collection, and the *Staphylococcus* study). This
resulted model has worse score metrics than VHIP (accuracy: 0.829, F1 score: 0.77, ROC: 0.89,
and Matthew's correlation: 0.639), which is expected as less data was used during the training
phase of the model. This inferior model was then applied on the data unused during training
and it predicted interactions with a 91% accuracy rate. This may suggest that data provenance
is not a significant driver of the accuracy of VHIP when trained on the entire dataset.

Comparing VHIP to existing host prediction tools is not straightforward. Existing HPTs
aim to identify the host for a given virus (i.e., answering the question "what taxa does this virus
infect?"). Whereas VHIP was designed to answer a fundamentally distinct question, where
given a virus and a list of hosts, the response is "whether infection is predicted to occur or not
occur for each host". To enable comparison, we challenged HPTs against known virus-host
pairs and recorded their top predictions. Since HPTs were trained on virus-host pairs
extracted from the virus NCBI database, we used virus-host pair associations from the Nahant
Collection and *Staphylococcus* study that were not used during training of VHIP, ensuring
novel data points across all tools. For each virus-host pair evaluation, we queried VHIP's infec-
tion prediction and checked if the correct host was included in each HPT's output. Out of 214
data points, VHIP correctly predicted infections 63.5% of the time. The accuracy of iPHoP,
VHMnet, vHULK, and CHERRY at predicting correct hosts was 0.93%, 4.2%, 4.2%, and 2.3%
respectively.

## Discussion

In this study, we make available the most comprehensive dataset of experimentally verified
virus-microbe interactions from publicly available databases and literature. This holistic
dataset allowed a reevaluation of assumptions about the prevalence of narrow versus broad
viral host ranges. We assessed taxonomic biases in virus-microbe culturing and found that the
lack of consistency in testing and reporting viral host range against a taxonomically diverse
panel of hosts may perpetuate the notion that viruses are specialists. Further, recent technolog-
ical advances and lab experiments are also challenging this conventional view of viral special-
ists, as well as revealing the diversity and complexity of virus-host interactions [17,18,55]. For
instance, the Hi-C metagenomic pipeline, which links DNA based on physical proximity
before sequencing, frequently results in highly nested networks where viruses' host ranges can

span broad taxa [56,57]. These metagenome-based findings are now routinely supported by host range experiments when a broader range of hosts is challenged [36–39,58]. Additional lab experiments testing viruses against diverse host panels and consistent reporting are necessary to ascertain the relative specificity versus breadth of viral host ranges.

Importantly, the dataset of experimentally verified virus-host microbe interactions compiled in this study allowed for the development of a new resource, VHIP, that deviates from existing virus-microbe prediction tools and opens a new avenue to the study of virus-host interaction networks. VHIP is distinct by design in that it predicts infection/non-infection for any given virus-host pair. This approach has multiple benefits. First, VHIP takes both viral'and putative host sequences as input, allowing a user to consider all viral and cellular populations recovered from a sampled community (Fig 6A). Second, VHIP may predict a virus to infect *multiple* different hosts, more accurately reflecting the nature of viral host ranges. Finally, VHIP can resolve complete virus-host interaction networks, which is only possible if a model can explicitly predict both positive and negative relationships between viruses and their potential hosts. Owing to these central design differences, it is impossible to fully compare the accuracy of VHIP with that of existing HPTs. This is because HPTs are designed to return a single host or a list of predicted hosts, and their accuracy calculation considers only the highest score. This is not a problem when the tools are trained on a dataset where there is only one known host, but this limitation becomes an issue when predicting hosts for a novel virus since the predictions are limited to the pool of taxa on which those models were trained. For VHIP, every virus-microbe pair combination is considered, such that the accuracy is defined by the ability of VHIP to accurately infer both infection and non-infection interactions.

The underlying assumption of predicting virus-microbe associations by leveraging genomic signals is that those signals are similar regardless of virus or host taxonomic assignment and/or environmental conditions (e.g., viral adaptation at the genome scale is similar the human gut microbiome to viruses in the oceans). If this assumption is violated, then one must be careful with interpreting predictions from sequence-based tools. To assess this issue, we used a subset of VHRnet containing a smaller number of virus-microbe pairs that contained an equal amount of data points from each data source (NCBI, Nahant Collection, and *Staphylococcus*). This model was then applied on the data not used for the training/testing phase of the model and obtained an 89% accuracy rate at predicting infection and non-infection events. Furthermore, when using the full dataset, VHRnet was divided into two sets: a set for the training/testing phase of the machine learning model, and a hold-out set to assess the model performance. Because variation can arise from how the data is divided between the two sets, this pipeline was bootstrapped 100 times. For each iteration, there was very little difference in the performance of VHIP (S7 and S8 Figs).

The perspective shift from predicting the most likely taxa a virus can infect to considering all possible virus-host pairs is necessary to resolve virus-host interaction networks (Fig 6A). Interaction networks are mathematical objects that capture and quantify the multitude of potential interactions between species, which provide a common framework for investigations across scales (Fig 6). In such networks, the nodes represent viral and host populations, and an edge connects a virus to a host if it is predicted to infect it. Additional data, sequence or otherwise, can be depicted with networks. Edges can be colored by properties of the interaction that depend on the unique combination of a given virus and host (e.g., viral fitness on a given host, whether infection is lysogenic or lytic) or by the phenotypic properties of the infected cell (virocell) during infection (Fig 6B). Population sizes could be represented by scaling the sizes of nodes, and frequency of infection could be encoded in the width of the edges (Fig 6C). By considering all possible virus-host pairs, such networks can be used to better understand microbial predator-prey interaction patterns and tease apart the underlying processes occurring across

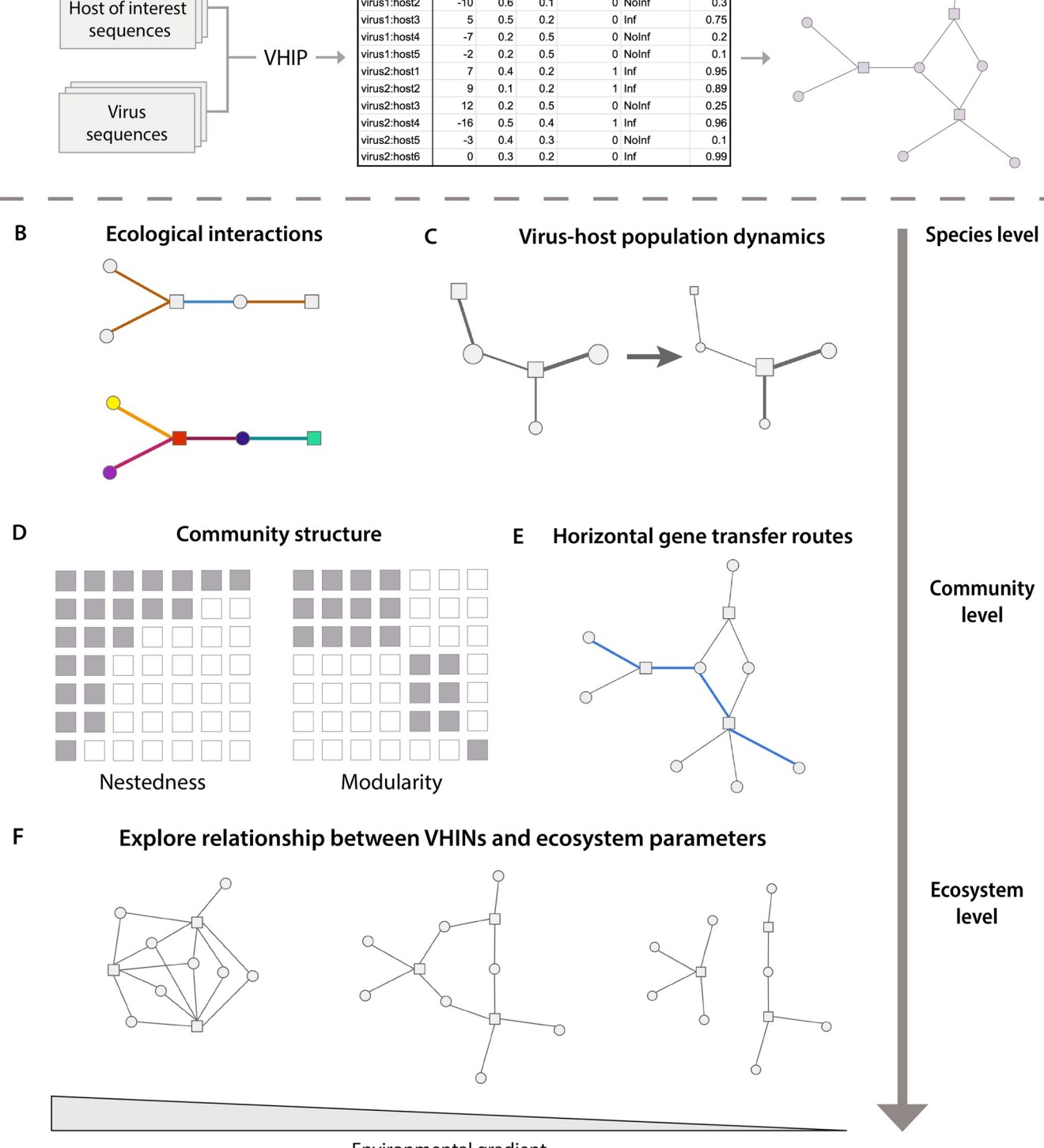

**Fig 6. Model output and future directions. A.** VHIP predicts interactions between all possible virus-host combinations from the input sequences of viruses and potential hosts. Because all possible pairs are considered, the VHIP predictions can be visualized as a bipartite network of virus-host interactions. **B.** Additional information describing the nodes (biological entities) and edges (interactions) can be mapped onto the network. **C.** Virus-host population dynamics can be coded and visualized in the nodes and edges. **D.** Community structure can be studied by considering the entire set of predicted interactions. **E.** Edges in the network can be thought of as different possible routes for horizontal gene transfer to occur. **F.** By comparing networks across space, time, and/or environmental gradients can help understand how virus-host interactions are structured and their underlying processes manifest at ecosystem scales.

scales [51,59,60](Fig 6D), inspiring hypothesis linking such structures and ecosystem properties. Further, virus-host networks explicitly model possible routes of infection-mediated horizontal gene transfer, facilitating the study of gene flow at both population and community scales (Fig 6E). At ecosystem scales, the predicted virus-host infection networks allow us to move beyond ecogenomics as the study of diversity across gradients and instead study how ecological interactions are structured across physicochemical, temporal, and spatial gradients. This shift allows for the integration of multilayer network theory into microbial ecology and opens new opportunities to study ecological complexity [59] (Fig 6F).

The application of network analyses permeates studies in ecology and evolution [61,62] and contributes to the understanding of community assembly [60,63,64], robustness and resilience [64,65], and species coexistence [66] across biological disciplines. Yet, the application of network thinking to the study of virus-microbe interaction is still in its infancy. The tool developed and presented here represents an important step with the power to leverage metagenomic data to answer the question "who infects whom?" from uncultured sequence data and supports new avenues to apply network thinking to viral ecology.

## Methods

### Collection of host range data and associated sequences

GenBank formatted viral genome files (n = 2621) were downloaded from the NCBI RefSeq database (Aug. 2018). Virus-host pairs were retrieved from two sources: (1) metadata fields in the viral genome GenBank file under the host or lab_host description, and (2) literature search reporting host range data (S1 Table). For the first source of host data, the 'lab_host' tag of each of the 2621 viral GenBank files was used to associate the virus with the host used in the sequencing project. If available, the host strain genome was downloaded from RefSeq (S5 Table). If a genome sequence for the host strain was not available, but a genome of the host species was sequenced, a representative genome of the host species was randomly chosen and downloaded from RefSeq (S5 Table). For the second source of host range data, the 'Title', 'Journal', and 'Author' tags of each viral GenBank file were used to identify primary journal articles (S1 Table). Lab-verified infection and non-infection data for the sequenced viruses were recorded from the identified reference articles. Additional studies that reported host range data for the sequenced viruses were identified via manual literature searches of the virus name. The data was compiled into a single file, named VHRnet for Virus-Host Range network (S2 Table).

### Comparison of existing virus-host prediction tools on complete host range experiments

The viral sequences belonging to the *Staphylococcus* study and the Nahant collection study were given as input for the following predictions tools using their default settings: VirHost-Matcher-Net (July 2021 version) [30], vHULK (v1.0.0) [31], CHERRY (v1.0.0) [27], iPHoP (v1.2.0) [29], and RaFAH (v0.3) [28]. To evaluate accuracy of those tools, the highest score prediction from each tool was considered. There are three possible outcomes: HPT correctly predicted a species that the virus can infect, HPT predicted a species that the virus cannot infect, and HPT predicted a host that was not tested experimentally. In cases where a virus could infect multiple different hosts, a tool only had to predict at least one host among the known hosts to be considered to have 100% accuracy. iPHoP differs from existing HPTs since it can return 0, 1, or multiple predicted hosts for a given virus. We considered separately the best prediction by iPHoP versus the set of hosts predicted by iPHoP when evaluating its performance

on the complete host range studies. For RaFAH, it can only return predictions at the genus level so no assessment of its accuracy at the species level was not possible. To calculate accuracy of those tools at higher taxonomic levels, we considered the taxonomic level of the highest score of the predicted species. For example, if a tool predicted *E. coli* as the most likely host, to determine the accuracy at the phylum level, the phylum of the known hosts were compared to the phylum of *E. coli*.

## Evaluation of commonly used features for virus-host predictions

The most commonly used features are sequence composition (i.e., how similar the pattern of *k*-mer frequencies between the viruses and their hosts are) and sequence homology (i.e., the presence of a DNA match between a virus and a host) (Fig 1C and 1D). The %G+C content was calculated for all the viral and host genomes using a custom Python script. The difference in %G+C content between viral and host genomes was defined as: viral%G+C—host%G+C. A custom Python script was used to generate *k*-mer profiles for the viruses and hosts using *k*-length of 3, 6, and 9, and to calculate similarities for each virus-host pair using the $d_2^*$ distance metric and the Euclidean distance metric.

Sequence homology, a stretch of DNA that matches between a virus and a host, was used to identify evidence of prior infection (e.g., in the form of remnant integrated prophages, horizontal gene transfer events, or CRISPR spacers). A BLASTn was run between all viral genomes belonging to VHRnet to the NCBI RefSeq (Aug. 2018) sequences database for bacteria and archaea (minimum identity percentage 80 and minimum length 500bp). CRISPR spacers were identified using the CRISPRCasFinder tool (v4.2.20) on all sequences of the NCBI RefSeq sequence database with the following settings: -keepAll -log -cas -ccvRep -rcfowce -getSummaryCasfinder -prokka -meta. Spacer sequences were extracted from the CRISPRCasFinder output. Since spacers are typically 30 to 35 nucleotides long, a BLASTn with the short setting flag was performed between viruses against spacers. Only virus-spacers hits with no mismatches were kept for the CRISPR feature. A Pearson pairwise correlation was performed to assess correlations between features (S3 Table).

## Comparison of machine learning classifiers using the VHRnet dataset

The signals of coevolution in combination with the knowledge of infection/non-infection for each pair constitute the input needed to explore machine learning model approaches. The input data was first randomly shuffled to ensure that any intrinsic non-random ordering of the data was removed and thus would not influence machine learning behavior. In addition, because the ratio of non-infection to infection in VHRnet is imbalanced (68.7 to 31.3), the host range data was first downsampled to reach a ratio of 60/40 of non-infection to infection. Different machine learning classifiers were tested using the scikit-learn package (v1.3) in Python (v3.10), namely AdaBoost, GradientBoostingClassifier, KNeighborsClassifier, RandomForest, StochasticDescentGradient, and SupportVectorClassifier (SVC). Each machine learning model has different values settings, herein referred as hyperparameters, that control the learning process during the training phase. The combination of hyperparameters that results in a robust model is both dependent on the training dataset and the features being considered. To determine the best performing machine learning model given the study design for virus-host predictions, we performed a broad grid search (using the GridSearchCV module from scikit-learn) to explore different combinations of hyperparameters.

During the grid search, 70% of the input data was used as the training/testing set and the remaining 30% was kept as a hold-out set. A shuffle split (n = 10) was used to divide the training/testing set into 10 splits, where 9 splits are used to train the model and the remaining one is used

to assess performance of the model. This is repeated until each split has been used as the test set. Once the best performing set of hyperparameters was determined using the training/testing set, it was then evaluated on the hold-out set (Fig 1E). This entire process, including the downsampling of non-infection data to obtain a 60/40 ratio of infection to non-infection events, was bootstrapped 50 times for each type of machine learning classifier, except for SVC for which a single grid search was performed due to the runtime required. Code and analysis of the grid search is available here: https://github.com/DuhaimeLab/VHIP_analyses_Bastien_et_al_2023

### Training, testing, and evaluation of VHIP

From the previous analysis, we determined that the Gradient Boosting Classifier performed best and therefore is the most appropriate model for virus-host predictions given our study design. We ran a more exhaustive grid search. During the grid search, the data was again downsample to reach a 60/40 ratio of non-infection to infection. For each iteration (n = 100), 70% of the host range data was used for training/testing of the mode, and the remaining 30% kept as hold-out to evaluate the best set of hyperparameters for that iteration (Fig 1E). In addition, when assessing the best combination of hyperparameters, the AUROC (S7 Fig) and F1 score (S8 Fig) were also computed to assess the model performance and consistency across each iteration. From this pipeline, we determined that the best combination of hyperparameters are: max_depth = 15, learning_rate = 0.75, and loss = exponential. Finally, the model was trained one more time using a shuffle split (n = 10), where 70% of the data was used for training and the remaining 30% for testing the model. The ROC, F1, and MCC scores were calculated using functions provided by the scikit-learn metrics module. Finally, the accuracy of the model was calculated as (TP + TN) / (TP + TN + FP + FN). VHIP is available through conda-forge and PyPI. The source code is available at: https://github.com/DuhaimeLab/VirusHostInteractionPredictor.

To assess the effect of data provenance on our machine learning model, the data was subsampled such that there were equal amounts of data points from each source (n = 3159). Because the subsampling can introduce randomness, this pipeline was bootstrapped 100 times. The accuracy, AUROC, F1 score, and Matthew's correlation were computed for each iteration of the model.

To compare VHIP's prediction ability to existing tools, we evaluated each tool in their ability to recover known infection virus-host pairs. For this assessment, only pairs from the Nahant Collection and *Staphylococcus* study in the test set were considered to ensure novel data points across all tools. For each virus-host pair evaluation, we queried VHIP's and checked if the correct host was included in each HPT's output.

### Supporting information

**S1 Fig. Distribution of all family classifications of viruses in VHRnet.** Lighter transparency represents the proportion of non-infection reports by viral family, relative to the solid portion, which represents known infection reports by viral family.
(TIF)

**S2 Fig. Distribution of all host genera represented in VHRnet.** Lighter color transparency represents the proportion of non-infection relative to infection (solid color).
(TIF)

**S3 Fig. Number of viruses tested against different host taxa.** X-axis represent the number of viruses that have been tested.
(TIF)

**S4 Fig. Kernel density of distance measurements for each virus-host pair, colored by interaction (yellow for infection and blue for non-infection).** The top row used the Euclidean distance to compute the similarity between the k-mer profiles of the virus and its host, while the second row uses the $d2^*$ distance metric. Each column represents a different length of k-mer used to create the k-mer profiles (k-length of 3 versus 6 versus 9). The $d2^*$ distance metric is a more appropriate metric than the Euclidean distance metric for the purpose of virus-host prediction since it encodes some evolutionary signals (the peaks for the no-infection and infection are separated).
(TIF)

**S5 Fig. Feature distribution (diagonal plots) and co-correlations (all the other plots).**
(TIF)

**S6 Fig. Comparison of different machine learning models on VHRnet.** For each type of machine learning model, a grid search was performed to determine the best combinations of parameters. This plot shows the accuracy of the best performing model. This was bootstrapped 50 times (except for SVM since the fit algorithm is $O(n^2)$).
(TIF)

**S7 Fig. ROC curves from 100 bootstrapping iterations of the best model trained during the grid search using best hyperparameters**
(TIF)

**S8 Fig. F1 curve of 100 best hyperparameter combinations during the grid search.**
(TIF)

**S1 Table. Compilation of NCBI accession numbers of lab-tested viral host range and their respective DOI.** Submitted as an excel spreadsheet.
(XLSX)

**S2 Table. Machine learning model input.** Each row contains an experimentally tested virus-host pair, their known interaction, and the signal of coevolutions computed from their genomic sequences. Submitted as an excel spreadsheet.
(CSV)

**S3 Table. Comparison between input and output of existing host-prediction tools.**
(XLSX)

**S4 Table. Pearson pairwise correlations of features that went into VHIP.** Higher value means the features are more strongly correlated.
(XLSX)

**S5 Table. List of NCBI accession numbers for viral and host sequences used in this study.** Submitted as an excel spreadsheet.
(CSV)

## Acknowledgments

We thank K. Shedden, B. Hegarty, and M. Moreno for helpful discussions and suggestions and Geoffrey Hannigan for early discussions and collaboration that inspired the hunt for more data.

## Author Contributions

**Conceptualization:** G. Eric Bastien, A. J. Wing, Melissa B. Duhaime.

**Data curation:** G. Eric Bastien, Rachel N. Cable, Cecelia Batterbee.

**Formal analysis:** G. Eric Bastien, Rachel N. Cable, Luis Zaman.

**Funding acquisition:** Melissa B. Duhaime.

**Investigation:** G. Eric Bastien, Cecelia Batterbee.

**Methodology:** G. Eric Bastien, Rachel N. Cable, Luis Zaman, Melissa B. Duhaime.

**Project administration:** Melissa B. Duhaime.

**Resources:** Melissa B. Duhaime.

**Software:** G. Eric Bastien.

**Validation:** G. Eric Bastien, A. J. Wing, Luis Zaman.

**Visualization:** G. Eric Bastien, Rachel N. Cable, Luis Zaman.

**Writing – original draft:** G. Eric Bastien, Melissa B. Duhaime.

**Writing – review & editing:** G. Eric Bastien, Luis Zaman, Melissa B. Duhaime.

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
