## [Decision Letter · Decision Letter 0]

19 Feb 2024

Dear Dr. Duhaime,

Thank you very much for submitting your manuscript "Virus-Host Interactions Predictor (VHIP): machine learning approach to resolve microbial virus-host interaction networks" for consideration at PLOS Computational Biology.

As with all papers reviewed by the journal, your manuscript was reviewed by members of the editorial board and by several independent reviewers. In light of the reviews (below this email), we would like to invite the resubmission of a significantly-revised version that takes into account the reviewers' comments.

In this work, the authors construct a dataset of viral host ranges by combining information from several publicly available sources. Using these data, they evaluate the performance of several existing tools for predicting viral/host interactions based sequence data and develop a novel method that leverages both sequence and other metadata. The reviewers all highlighted the potential utility of the data set, which I agree with, but also identified several areas where substantive additional work is needed.

First, the reviewers all agreed that the biased nature of the host/viral data wrt to host limited the impact of both the new dataset and our interpretation of the novel machine learning method. My suggestion is that the authors demonstrate more rigorously how model performance varies when considering the smaller number of non-human pathogenic viruses included in the study. For example, the authors could split training/testing data between human pathogenic and all other viruses or could do a kind of repeated bootstrap or downsampling such that human pathogenic vs. non-human pathogenic sample sizes are equal. Additionally, the authors should further discuss what might be different wrt to model performance, etc. if more diverse data sets were available.

Second, the reviewers point out that while a comparison is done between existing classification methods, no comparison seems to have been done between the new developed method and established approaches. Given how much emphasis in the paper is placed on both the specific new method and, more interestingly, the general approach of including more metadata in the prediction models, the authors must provide a meaningful demonstration that their approach is in fact better. Importantly, the authors must also engage data set bias when comparing across models. I can imagine a number of pernicious kinds of bias that could show up here related to metadata coverage in human vs. non-human pathogenic viruses, etc.

Third, a number of claims in the paper, e.g., the network results in F6, seem to go well beyond what is demonstrated in the paper. My advice here is to carefully weigh what is and isn't a well supported conclusion in the current study. For example, given the heavy reliance on two data sets, it does seem unlikely that much can be said by generalizability across environments.

I hope that the authors pay careful attention to the detailed reviewer comments and understand that addressing them will almost certainly require running new analysis and a meaningful re-write of aspects of the manuscript.

We cannot make any decision about publication until we have seen the revised manuscript and your response to the reviewers' comments. Your revised manuscript is also likely to be sent to reviewers for further evaluation.

Sincerely,

Samuel V. Scarpino

Academic Editor

PLOS Computational Biology

James O'Dwyer

Section Editor

PLOS Computational Biology

In this work, the authors construct a dataset of viral host ranges by combining information from several publicly available sources. Using these data, they evaluate the performance of several existing tools for predicting viral/host interactions based sequence data and develop a novel method that leverages both sequence and other metadata. The reviewers all highlighted the potential utility of the data set, which I agree with, but also identified several areas where substantive additional work is needed.

First, the reviewers all agreed that the biased nature of the host/viral data wrt to host limited the impact of both the new dataset and our interpretation of the novel machine learning method. My suggestion is that the authors demonstrate more rigorously how model performance varies when considering the smaller number of non-human pathogenic viruses included in the study. For example, the authors could split training/testing data between human pathogenic and all other viruses or could do a kind of repeated bootstrap or downsampling such that human pathogenic vs. non-human pathogenic sample sizes are equal. Additionally, the authors should further discuss what might be different wrt to model performance, etc. if more diverse data sets were available.

Second, the reviewers point out that while a comparison is done between existing classification methods, no comparison seems to have been done between the new developed method and established approaches. Given how much emphasis in the paper is placed on both the specific new method and, more interestingly, the general approach of including more metadata in the prediction models, the authors must provide a meaningful demonstration that their approach is in fact better. Importantly, the authors must also engage data set bias when comparing across models. I can imagine a number of pernicious kinds of bias that could show up here related to metadata coverage in human vs. non-human pathogenic viruses, etc.

Third, a number of claims in the paper, e.g., the network results in F6, seem to go well beyond what is demonstrated in the paper. My advice here is to carefully weigh what is and isn't a well supported conclusion in the current study. For example, given the heavy reliance on two data sets, it does seem unlikely that much can be said by generalizability across environments.

I hope that the authors pay careful attention to the detailed reviewer comments and understand that addressing them will almost certainly require running new analysis and a meaningful re-write of aspects of the manuscript.

Reviewer's Responses to Questions

**Comments to the Authors:**

Reviewer #1: I do not have any major issues with the manuscript.

As a suggestion, please consider testing the tool on genomes derived from metagenomes. In particular ones with organisms that are taxonomically distanced from the ones dominating your existing host range datasets. There are some metagenomic datasets that include both long read data and Hi-C to confirm. I believe https://doi.org/10.1016/j.jhazmat.2023.131944 - which you cite - has some. My main concern is the relatively narrow host range your tool was developed on. Will the model generalize?

Also, for future work, consider using graph neural networks once you get to ecosystems.

Reviewer #2: In this study, Bastien et al. manually curated a database of virus-host relationship based on publicly available experimental data. Both infection and non-infection information, as well as many-to-many infectious relationships were included, which is distinguishable from many known virus-host databases. Based on this database, the accuracies of commonly used host prediction tolls were validated. The authors then evaluated genomic features between viruses and hosts resulted from coevolution, and based on selected features, they developed a machine-learning based classifier to predict virus-host infectious features based on their genome sequences.

The concept of including non-infection information to host prediction, the attempt in resolving all the infection/non-infection relationships in a community of viruses and hosts as emphasized by the authors are indeed very important. However, they have not clearly demonstrated that the prediction tool they developed has benefitted from these additional data in performance. Moreover, the authors need to provide a user-friendly implementation of this tool online.

Major concerns.

(1) As the authors have admitted, the VHRnet database is heavily biased to human pathogens, phage therapy studies, and the data from the two experimental studies they utilized. There are so many potential relationships between the hosts and viruses in the database are not yet tested by experiments. Therefore, its current version can provide little valuable information to assess the topic of host specificity of viruses. Please move lines 164 to 175 to the Discussion, and instead of indicating that your database indicates that viruses are specialists, you should point out that the VHRnet database currently may provide little valuable information to host specificity of viruses.

(2) The authors explained why they did not compare the performance of their tool to other published host prediction tools with a very vague description. They seem to say that VHIP is designed for predict virus-host pair combination while all others predict hosts based on highest scores? I am not convinced. I think you can certainly compare the accuracy of VHIP with others for any single host-virus pairs as you have done for the existing tools in Fig. 3. The authors reported an overall accuracy value of 87.8%, but it is not explained how this value is obtained? I can not found relevant figures or tables for the results of this calculation.

(3) I checked the github website of VHIP, but it looks like that VHIP is not yet fully implemented in a user-friendly way. The pretreatments for data needs to be done by the users with additional tools and procedures, which lacks of necessary details in introduction. This exclude most of the users without rich experience in bioinformatics. Please fully implement VHIP as a conda package or at least easy to use. Also please provide a detailed instruction for installation and use of this software.

(4) The discussion of virus-host networking modeling is way beyond the currently results can infer. Please delete Fig. 6 and most of the relevant discussion, unless you can provide the results showing that you have applied VHIP on a specific environmental data to build and explore a network. Instead, please discuss more about the accuracy and reliability of your tool.

Minor comments.

Line 132. Legends of C and D should be swapped.

Line 235. Methods to identify HGT is not found, please specify the sequence identity threshold here.

Line 364. a stronger signal by compared to what?

Fig 5D. Please order by the one with highest importance on the top and the ones with lower importance following up.

Reviewer #3: Overall thoughts

This paper trains a Gradient Boosted Machine to predict the presence of a successful interaction between viruses and bacterial hosts using genome features as predcitors. While the creation of these genome features is outside of my immediate expertise, I found they employed a nice use of CRISPR motifs in host and virus genomes as a signal of past co-evolution, even though they found that these motifs were rare in their data. The authors conclude that features including GC differences and k-mer sequence motifs were the most useful features.

While I think this paper is an excellent forway into the use of both virus and host genome features to predict virus-host interactions, I found that the paper largely over-sold it’s novelty in terms of methodology, and the production of a new and valuable dataset of experimentally verified infections. In particular, the authors claim novelty of this paradigm of binary network prediction for species interactions, which indicates they did not conduct a thorough literature review. Further, their dataset is largely an amalgamation of two existing datasets, which are narrow either taxnomically or sample a single environment. They do add new virus-host interactions to these published records, but they are mined from GenBank metadata without any discussion of validation of these records, or why they assume them to be controlled experimental infections as opposed to naturally infected hosts. From a methods perspective, despite acknowledging their limited and unbalanced training data, the authors do not explore any methods for balanced sampling / re-sampling / data augmentation to help these imbalance issues.

Finally, while the authors provide a nice general discussion, they do not provide any ecological or evolutionary context for the specific predictions made by their model. This would greatly improve the general interest of the work, and also help to open the “black box” of their model.

I’m sorry I cannot be more positive. I think this has the potential to be a very good paper if more attention is paid to the data and modelling, and the claims of novelty and applicability are reduced, and the model and results are placed in a better context with respect to the authors study system (bacteriophages), and the broader landscape of species interaction models.

Major comments

Insufficient review of literature and overall context for the current work:

Lines 40-43 & 82-93: There are multiple virus-host prediction models that predict muliple interactions at once (e.g. the entire network or parts of the sub-network). Further, many existing models successfully reconstuct multiple features of these networks, and in many cases the accuracy of these models excees that reported in your abstract (87.8%) From these statements and the cited references, it is clear that the authors have not sufficiently reviewed existing literature / methods in this field. For example, relevant literature includes:

- Albery et al. (2021) The Science of the host-virus network. Nature Microbiology

- Wardeh et al. (2021) Divide-and-conquer: machine-learning integrates mammalian and viral traits with network features to predict virus-mammal associations. Nature Communications

- Farrell et al. (2022) Predicting missing links in global host–parasite networks. Journal of Animal Ecology

- Elmasri et al. (2020) A hierarchical Bayesian model for predicting ecological interactions using scaled evolutionary relationships. Annals of Applied Statistics

- Poisot et al. (2023) Network embedding unveils the hidden interactions in the mammalian virome. Patterns

- Strydom et al. (2023) Graph embedding and transfer learning can help predict potential species interaction networks despite data limitations. Methods in Ecology & Evolution

I strongly suggest the authors conduct a mor thorough review of alternative approaches for host-pathogen prediction, and adjust their claims accordingly. In particular, in Figure 6 you say your predictions can be visualized as a bipartite network of all virus-host interactions. There is a wealth of information and models on bipartite network models which could help to inform your approach.

Concerns about the newly assembled dataset:

Collecting lab-verified interactions is an excellent way to train and validate such a model, especially if you have “true negatives”, which are often absent from many databases of host-virus interactions. When presenting your dataset, it would be great to indicate what evidence you considered as proof of a successful infection. For example, if you found the same virus-host interaction tested in two studies, one successful, another not, how would you code this? Would you discard the unsuccessful experiment, or does your model allow for both of these observations as input?

In terms of the provenance of the input data for “VHRnet”, for the NCBI data it is unclear how “clean” or reliable the host tage in GenBank is. Having worked with these data, I have found multiple entires where this information is incorrect. What steps did you take to verify these data? Even manual curation of a small random subset would be a good attempt. Further, if you are using RefSeq, how do you determine if the sequence comes from a lab controll experimental infection rather than an observational study (e.g. virus may be present as a contaminant), and in this context what does a “non-infection” study look like in terms of the GenBank meta-data? For example, in lines 151-152 you say the majority of viruses were reportedly tested again a single host, and these pairings come from the “host” tag, however the host tag could be the source from which the virus was sampled and a genome was assembled from metagenomic data, but this does not tell you if this was an experimental infection or not.

It seems like the majority of your data come from the Nahant study. This database appears to be sampled from one ecosystem (a littoral marine zone). Further, there is no discussion of whether the three sources you used are comparable in terms of the definition of a successful infection, or how this was assessed. Considering over 70% of your virus-host pairs come from this study, and another on Staphylococcus, and you focus extensively on these two datasets in your results, I wonder if it is appropriate to say you have created a new database of lab-tested host-virus interactions. Further, this raises the question of how confident can you be with your predictions as they extend to hosts in other systems (e.g. human microbiome, phyllosphere, soil microbiome, animal microbiomes, etc...)?

Also a small semantic point as mentioned above, this database is actually more of an edge list (list of host-virus interactions, along with their successful experimental infection).

Issues with the modelling appraoch

Line 196-197: You say that existing HPTs performed better in predicting hosts for viruses in the Staph study than viruses in Nahant. Could this be due to an imbalance in your training data? E.g. You have more data for Staph viruses, hence are less able to generalize to other host groups? Considering the potential data quailty and imbalance issues I would have expected some exploration of data re-sampling and/or attempting to fit your model on one dataset and then predict the other.

Lines 329-336: You should be able to assess whether models are overfit by testing on a hold-out / validation test set. In my experience, models are picked based on performance (assessed via multiple metrics) on validation set, rather than that number of trees. This also limits your ability to compare models which do not have this underlying architecture.

Line 336: What is the “untouched dataset”? This should be defined earlier.

Lines 329-361: Much of this reads like Methods rather than results. It would be good to see more results in the context of predictive accuracy across different taxonomic groups, as a function of the input data (which of the data subsets the models were trained on), and discussing some examples of when the model is successful, and then it fails. By examining which interactions your model is predicting as false positives and false negatives you may be able to get some useful insights into why it may fail, and how to build a better next iteration of your model.

Figure 6: It is unclear how your model extrapolates predictions from the species through to the community and ecosystem levels.

Minor comments

Lines 30-31 & 69-70: Please provide more context for “Advances in genome sequencing have led to the discovery of millions of novel viruses” – it is unclear which species concepts you are applying here, as there are fewer than 7,000 recognized virus species, and while I am aware of studies using different forms of extrapolation and power laws for estimating millions of viruses, these are quite variable and somewhat contested...

Lines 32 & 34 & 79: Stylistic comment, but I think “what” is more appropriate than “who” when talking about bacteria and viruses.

Line 38: “host range” is often used when discussing the diversity of hosts a pathogen can infect (e.g. how many hosts / species richness, or perhaps phylogenetic diversity, etc.), as a reference to specialism and generalism in terms of host tropism, rather than the particular host-pathogen interaction. Unless you are explicitly predicting host range, using this phrase is incorrect.

Line 40: Unclear what “features of co-evolution” are.

Lines 45 & 76: Unlear what you mean by “popuation genomes”. Please define this and differentiate from genomes typically associated with individual species.

Line 82: Is “host prediction tools (HPT)” a commonly used term in this field? I am more familiar with host-virus link prediction models / network models, etc...

Line 91-93: I don’t understand the idea that these models “do not predict non-infection”. Aren’t most of these models assuming a binary outcome, and thus predicting an interaction for some host-virus paris also by necessity predicting the absence of a similar interaction in a different host-virus pair? If HPTs are very different than existing network prediction / species interaction / link prediction models, it is important to offer a more detailed summary of these approaches and how they differ.

Lines 101-103: Given the format of results then methods, it would greatly help the reader if you have a breif overview of the types of co-evolutionary signals you measured (are these identified sites of selection, or auto-generated genome composition features?), and the general type of ML model employed (GBM) somewhere here in the introduction.

Lines 110-112: Please clarify that these are bacteriophages and bacteria, correct? If so, these are not all viruses and hosts as you exclude large taxonomic groups of viral hosts here.

Line 308-312: What cutoff for correlation did you use? Depending on the ML model used, some relatively “highly correlated” predictors could provide very useful information. For example, in GBMs, which allow for highly non-linear interactions between predictors, predictors which appear fairly correllated could actually provide important discriminatory information for particular classes.

Line 352: You first use AUROC, then here switch to ROC when referring to AUROC.

**Have the authors made all data and (if applicable) computational code underlying the findings in their manuscript fully available?**

Reviewer #1: Yes

Reviewer #2: Yes

Reviewer #3: Yes

PLOS authors have the option to publish the peer review history of their article (what does this mean?). If published, this will include your full peer review and any attached files.

Reviewer #1: No

Reviewer #2: **Yes: **Lu Fan

Reviewer #3: No
---

## [Decision Letter · Decision Letter 1]

13 Jul 2024

Dear Dr. Duhaime,

Thank you very much for submitting your manuscript "Virus-Host Interactions Predictor (VHIP): machine learning approach to resolve microbial virus-host interaction networks" for consideration at PLOS Computational Biology. As with all papers reviewed by the journal, your manuscript was reviewed by members of the editorial board and by several independent reviewers. The reviewers appreciated the attention to an important topic. Based on the reviews, we are likely to accept this manuscript for publication, providing that you modify the manuscript according to the review recommendations.

I agree with both reviewers that the manuscript has improved substantially during revision. I would ask the authors to provide the comparisons requested by reviewer 2 and to ensure that the points raised by reviewer 3 in their original assessment are discussed in the manuscript (in addition to the response letter). I leave the decision around whether to keep figure 6, move it to the supplement, or save it for a future paper to the authors. However, I agree with the sentiment of reviewer 2 that the figure doesn't add much and is pretty high-level.

Sincerely,

Samuel V. Scarpino

Academic Editor

PLOS Computational Biology

James O'Dwyer

Section Editor

PLOS Computational Biology

I agree with both reviewers that the manuscript has improved substantially during revision. I would ask the authors to provide the comparisons requested by reviewer 2 and to ensure that the points raised by reviewer 3 in their original assessment are discussed in the manuscript (in addition to the response letter). I leave the decision around whether to keep figure 6, move it to the supplement, or save it for a future paper to the authors. However, I agree with the sentiment of reviewer 2 that the figure doesn't add much and is pretty high-level.

Reviewer's Responses to Questions

**Comments to the Authors:**

Reviewer #2: Thank you for your response to my comments and the changes you have made in the revision. I do have two further comments regarding to your response.

To your response to my 2nd major comment:

I think the author can still compare the accuracy of VHIP with existing tools 1) based on the conventional metric (e.g. either prediction of species A or species B is considered 100% accurate), and 2) based a new metric (e.g. only prediction of both species A and B is considered 100% accurate). In the latter case, some existing tools may still be applicable since they can make multi-host prediction based on a threshold cutoff of confidence score. I think both metrics have applicable scenarios in virus-host studies.

To your response to my 4th major comment:

I still think Fig. 6 is immature and it breaks the integrity of this manuscript. Please save it for your next paper.

Reviewer #3: I reviewed a previous version of this manuscript. The authors have addressed some of my concerns, including exploring potentially biased data through sub-sampling, and stating why some classes of link prediction models developed for vertebrate host-virusses (e.g. phylogeographic models) may not be applicable to predict microbe-virus interactions.

For my other concerns, I find that the authors address these in the response letter but it is unclear that they have made appropriate changes in the manuscript (e.g. viral species concepts / vOTUS are not mentioned, “features of co-evolution” are still mentioned and genomic features are not explicitly linked to co-evolutionary mechanisms, and it is unclear why HPTs are different than host-virus link prediction models).

**Have the authors made all data and (if applicable) computational code underlying the findings in their manuscript fully available?**

Reviewer #2: Yes

Reviewer #3: Yes

PLOS authors have the option to publish the peer review history of their article (what does this mean?). If published, this will include your full peer review and any attached files.

Reviewer #2: No

Reviewer #3: No

Figure Files:

Data Requirements:

Reproducibility:

References:

---

## [Editor Report · Decision Letter 2]

2 Sep 2024

Dear Dr. Duhaime,

We are pleased to inform you that your manuscript 'Virus-Host Interactions Predictor (VHIP): machine learning approach to resolve microbial virus-host interaction networks' has been provisionally accepted for publication in PLOS Computational Biology.

Best regards,

Samuel V. Scarpino

Academic Editor

PLOS Computational Biology

James O'Dwyer

Section Editor

PLOS Computational Biology

---

## [Editor Report · Acceptance letter]

12 Sep 2024

PCOMPBIOL-D-23-01780R2 

Virus-Host Interactions Predictor (VHIP): machine learning approach to resolve microbial virus-host interaction networks

Dear Dr Duhaime,

I am pleased to inform you that your manuscript has been formally accepted for publication in PLOS Computational Biology. Your manuscript is now with our production department and you will be notified of the publication date in due course.

With kind regards,

Jazmin Toth
